# Evaluation of Growth and Production of High-Value-Added Metabolites in *Scenedesmus quadricauda* and *Chlorella vulgaris* Grown on Crude Glycerol under Heterotrophic and Mixotrophic Conditions Using Monochromatic Light-Emitting Diodes (LEDs)

**DOI:** 10.3390/foods12163068

**Published:** 2023-08-16

**Authors:** Evagelina Korozi, Io Kefalogianni, Vasiliki Tsagou, Iordanis Chatzipavlidis, Giorgos Markou, Anthi Karnaouri

**Affiliations:** 1Laboratory of General and Agricultural Microbiology, Department of Crop Science, Agricultural University of Athens, Iera Odos 75, 11855 Athens, Greece; evangeliakorozi@yahoo.com (E.K.); bmic7kei@aua.gr (I.K.); tsagouv@hotmail.com (V.T.); chatzipavlidis@aua.gr (I.C.); 2Laboratory of Food Biotechnology and Recycling of Agricultural By-Products, Institute of Technology of Agricultural Products, Hellenic Agricultural Organization-Demeter, Leof. Sofokli Venizelou 1, Lykovrysi, 14123 Athens, Greece

**Keywords:** microalgae, crude glycerol, mixotrophy, heterotrophy, monochromatic light-emitting diodes (LEDs), lipids, proteins

## Abstract

This study aimed to examine the impact of crude glycerol as the main carbon source on the growth, cell morphology, and production of high-value-added metabolites of two microalgal species, namely *Chlorella vulgaris* and *Scenedesmus quadricauda*, under heterotrophic and mixotrophic conditions, using monochromatic illumination from light-emitting diodes (LEDs) emitting blue, red, yellow, and white (control) light. The findings indicated that both microalgae strains exhibited higher biomass yield on the mixotrophic growth system when compared to the heterotrophic one, while *S. quadricauda* generally performed better than *C. vulgaris*. In mixotrophic mode, the use of different monochromatic illumination affected biomass production differently on both strains. In *S. quadricauda*, growth rate was higher under red light (μ_max_ = 0.89 d^−1^), while the highest biomass concentration and yield per gram of consumed glycerol were achieved under yellow light, reaching 1.86 g/L and Y_x/s_ = 0.18, respectively. On the other hand, *C. vulgaris* demonstrated a higher growth rate on blue light (μ_max_ = 0.45 d^−1^) and a higher biomass production on white (control) lighting (1.34 g/L). Regarding the production of metabolites, higher yields were achieved during mixotrophic mode in both strains. In *C. vulgaris*, the highest lipid (26.5% of dry cell weight), protein (63%), and carbohydrate (20.3%) contents were obtained under blue, red, and yellow light, respectively, thus indicating that different light wavelengths probably activate different metabolic pathways. Similar results were obtained for *S. quadricauda* with red light leading to higher lipid content, while white lighting caused higher production of proteins and carbohydrates. Overall, the study demonstrated the potential of utilizing crude glycerol as a carbon source for the growth and metabolite production of microalgae and, furthermore, revealed that the strains’ behavior varied depending on lighting conditions.

## 1. Introduction

In recent years, there has been a lot of research and economic interest in developing biotechnological processes with microalgae for the production of high-added value products [1]. Microalgal biomass is considered an important renewable source, with many promising applications regarding the synthesis of a great repertoire of important compounds that can be employed in aquaculture feeds, as human dietary supplements and cosmeceuticals, as well as in the pharmaceutical industry [2]. Microalgae species appear to be extremely diverse and heterogeneous; nevertheless, the species that are widely used in the industry have to be resilient, highly productive [3], or capable of accumulating high lipid content that is extremely appealing for biodiesel production or as nutraceuticals [4]. Photoautotrophic microalgae are industrially important microorganisms, not only due to their ability to convert light energy to biomass and valuable metabolic products but also because they exhibit exceptional metabolic flexibility, which allows them to grow also in a variety of trophic modes including heterotrophy and mixotrophy [5]. In the heterotrophic mode, microalgae use organic molecules present in the media, such as glucose, glycerol, acetate, etc., to meet their energy and carbon demands for growth. On the other hand, in the mixotrophic mode, the microalgae maintain their capacity for photosynthesis while simultaneously using the organic substrate as a source of carbon and/or energy [6]. Although biomass and lipid yields have been reported to be significantly higher under heterotrophic or mixotrophic conditions compared to autotrophic growth, organic carbon sources are prohibitively expensive on an industrial scale compared to the other nutrients used. Due to the fact that crude glycerol is easily made available as a byproduct of biodiesel production, it is considered to be a promising source of organic carbon for the cultivation of microalgae to overcome the obstacle of high costs of fermentation substrate [7].

Both in the autotrophic and mixotrophic mode of cultivation, the light intensity and wavelength have been proven to hold a key role in affecting the cell growth rate, the metabolic processes and the biochemical composition of the microalgae [5]. Since different wavelengths induce modifications in the metabolic pathways of microalgae, a number of studies focused on the effects of artificial lighting with light-emitting diodes (LEDs) on the growth rate of various microalgal strains. Compared to conventional lighting (such as fluorescent bulbs), LEDs are able to provide microalgae cultures with an energy-efficient light source. Their key role is attributed to the fact that they are able to produce light to a wavelength spectrum that promotes specific photosynthetic reactions [8]. Microalgae require a specific wavelength that is optimum for the cells in order to achieve the maximal rate of photosynthesis with a lower energy cost. Up to now, research focused on reducing energy consumption and, consequently, the utilization of LEDs as a light source; *Auxenochlorella protothecoides* [8] *Spirulina platensis* [9], *Dunaliella salina* [10], *Nannochloropsis oceanica* CY2 [11], and mixed cultures of the *Chlorella* species [12], along with other microalgae, have been studied up to this point. The wavelength (spectrum) and intensity (quantity) are parameters that regulate the growth rate of microalgae by influencing the photosynthetic process. Microalgae are able to grow outdoors as autotrophic microorganisms in artificially created or natural ponds, although they will eventually suffer from low photosynthetic efficiency due to inhomogeneous light distribution between different parts of the culture [13]. Thus, in order to achieve the highest growth rates and photosynthetic effectiveness, the industry is focusing its efforts on optimizing microalgae growth conditions. Given that not all light wavelengths have the same impact on the photosynthetic mechanism, the photosynthetic process is primarily dependent on the light spectrum. Considering that photosynthetic pigments absorb different light wavelengths, microalgae respond in distinct ways under varying illumination conditions. Chlorophyll a is the primary active pigment in the photosynthetic reaction, while porphyrin pigments, such as chlorophyll b, c, and d, play a supporting role in light absorption, while several microalgal species also have the ability to synthesize astaxanthin, lutein, beta-carotene, and other carotenoid pigments, which are of high value [14]. Currently, research findings have proved that, besides growth, many other biochemical processes of microalgal cells, including the accumulation of proteins, lipids, carbohydrates, and pigments, are influenced by the quality of light [15,16]. LED lighting is a promising new method that can bring new insights into the implications that different wavelengths impact on the metabolic processes of microalgae. However, until recently, little research has focused on the effect of varied monochromatic lighting on the growth and manipulation of microalgae in mixotrophic conditions.

The current study attempted to evaluate the growth rate, biomass yield, cell morphology, and production of high-value metabolites (proteins, lipids, and carbohydrates) of the microalgae *Scenedesmus quadricauda* and *Chlorella vulgaris* when grown on crude glycerol as a carbon source on different trophic conditions. Both species are thought to be promising since they are capable of utilizing glycerol as an alternate carbon source [7] and could also be exploited as food and feed supplements [17]. Several strains of *C. vulgaris* and *Scenedesmus* sp. have been studied for growth and metabolite production on substrates with crude glycerol as a carbon source both heterotrophically and mixotrophically [7,18,19,20,21,22,23]. However, *S. quadricauda*, although a promising oleaginous microalgae that can accumulate lipids up to 33.1% of cell dry weight in mixotrophic culture [24], has not been investigated for its ability to grow on crude glycerol. By determination of growth patterns and cell composition in mixotrophic cultivations under different wavelengths of the light source and comparison with heterotrophic conditions, this work expects to provide novel insights on the optimization of culture parameters, as well as to shed light on the accumulation of valuable metabolites in microalgal cells. To the best of our knowledge, this is the first study that reports a point-to-point comparison of heterotrophic and mixotrophic growth of *S. quadricauda* and *C. vulgaris* on crude glycerol and, moreover, examines the effect of different lighting conditions on the cell growth and metabolism.

## 2. Materials and Methods

### 2.1. Microorganisms and Culture Conditions

*S. quadricauda* (CCAP 276/21) and *C. vulgaris* (CCAP211/51) were obtained from the Culture Collection of Algae and Protozoa SAMS Limited Scottish Marine Institute, Dunberg, Oban, Scotland, United Kingdom.

Stock cultures were maintained in Bold’s Basal Medium (BBM) supplemented with 10 g/L glucose and 2 g/L peptone. BMM consisted of 0.075 g/L KH_2_PO_4_, 25 mg/L CaCl_2_, 0.175 g/L K_2_HPO_4_, 0.075 g/L MgSO_4_·7H_2_O, 0.025 NaCl g/L, 10 mg/L NaFeEDTA, and 1.0 mL/L of the trace elements stock solution. The composition of the trace elements stock solution was 8.82 g/L ZnSO_4_·7H_2_O, 1.44 g/L MnCl_2_, 0.71 g/L MoO_3_, 1.57 g/L CuSO_4_·5H_2_O, and 0.49 g/L Co(NO_3_)_2_·6H_2_O. All cultures were axenic and were handled under aseptic conditions. Pre-cultures were grown in BBM supplemented with 10 g/L glucose and 2 g/L peptone on an orbital shaker set to 150 rpm, for 5 days, at 25 °C, under 24 h cool white LED panel illumination (100 μmol/m^2^/s) and were used to inoculate the main cultures, with BBM medium containing crude glycerol as the carbon source. Inoculum constituted 20% of the main culture on a volume basis. All experiments were conducted in 500 mL Duran flasks with a working volume of 300 mL. Aeration and adequate agitation were provided by pumping air through a 0.2 μm sterilized filter. The temperature was kept constant at 25 °C and cultivation lasted for 12 days.

Both heterotrophic and mixotrophic cultures were performed on BBM supplemented with 10 g/L of crude glycerol. Crude glycerol was obtained from P.N. PETTAS S.A., Patras, Greece, with the following composition: glycerol (94.1–95.1 wt%), ash (0.16–0.35% wt%), methanol (0.04–0.15% wt%), moisture (1.3–2.5%), and MONG (Matter Organic Non-Glycerol; 2.5–3.56% wt%). Peptone was selected as the nitrogen source at a concentration corresponding to a C/N ratio of 20:1. Heterotrophic cultures were run under totally dark conditions (in a dark chamber) in a batch mode, until they reached the stationary phase. In the case of the mixotrophic cultures, illumination was provided by five LEDs (SMD type; 14.4 W per meter, 60 SMD LEDs per meter; GloboStar LED Lighting Group, Katerini, Greece) per flask. The illumination for each culture flask was provided from the bottom by 5 LEDs shared in two LED stripes (total stripes length of around 8.5 cm). The flasks were illuminated from their bottom at a photosynthetic photon flux density (PPFD) of 105 ± 5 μmol/m^2^/s and were covered with aluminum foil to eliminate the loss of light irradiation and to prevent the penetration of other wavelengths. Four different monochromatic colors of illumination were used, namely white, blue, red, and yellow, while the white light source was used as control. The emission wavelength peak for each light color was 440 nm (for blue), 590 nm (for yellow) 630 nm (for red), as measured with SpectraPen LM 510. The photoperiod was set at 16 h light and 8 h dark. Photon flux was measured with a SpectraPen LM 510 (Photon Systems Instruments, Drásov, Czech Republic). All the experiments were conducted in triplicates.

### 2.2. Evaluation of Cells Morphology

Samples were taken from the cultures on a daily basis to evaluate the cell morphology and detect any possible bacterial contamination (no contamination was observed during the cultivation). The size and the morphology of the microalgae during stationary phase was determined at 100× magnification using an optical microscope. Each observation was based on the measurement of 50 cells.

### 2.3. Analytical Methods

Microalgal cells were harvested at the stationary phase of the culture through centrifugation (13,000 rpm for 10 min), washed twice with distilled water, and dried overnight at 80 °C. The biomass yield was assessed gravimetrically by estimating the dry weight. The residual glycerol in the culture medium at the end of cultivation was assessed by using kit from Megazyme (Ireland), following the instructions of the manufacturer.

Cellular total lipids were extracted from dried cells with a chloroform: methanol: water solvent system at a ratio of 2:1:0.2 and their concentration was estimated using a modification of the sulfo–vanillin method [25]. Briefly, 20 μL of the sample’s chloroform phase, which contained lipids in a concentration range of 200–500 mg/L, were incubated at 80 °C in a water bath to evaporate the solvent. At the following step, 0.4 mL of 96% sulfuric acid was added followed by boiling for 10 min. The samples were then left to cool at room temperature and 1 mL of vanillin/phosphoric acid stock solution was added. The solution stock was made by dissolving 0.12 g of vanillin in 20 mL DI water, followed by 80 mL of 85% phosphoric acid. Incubation took place for 15 min at 37 °C and the optical density (OD) of the samples was measured at 530 nm. For the construction of the standard curve, canola oil was used.

The carbohydrate content was determined using a modified phenol-sulfuric method [26]. Briefly, 10 μL of 90% phenol solution was mixed with 0.5 mL of centrifuged cell sample containing 10–50 mg/L carbohydrates. Then, 1.25 mL of 96% sulfuric acid was added and mixed well. The samples were left at room temperature for 30 min and then OD was measured at 485 nm. The carbohydrate content was determined using a standard curve with D-glucose.

Cellular proteins were extracted from biomass cells upon addition of 0.5 N NaOH and incubation for 20 min at 100 °C. The protein content was then estimated according to the method described by [27]. In brief, 100 μL of the sample containing the extracted proteins were mixed with an equal quantity of a 5% *w*/*v* SDS solution and 1 mL of a solution of 2% *w*/*v* Na_2_CO_3_ in 0.1 N NaOH. The samples were then left for 15 min at room temperature. Subsequently, a volume of 100 μL of freshly prepared 1 N Folin–Ciocalteu reagent (Penta, Chemicals Unlimited, Prague, Czech Republic) was added. The samples were incubated for 30 min in the dark at room temperature and then OD was measured at 750 nm. Bovine serum albumin was used for the standard curve.

For the determination of chlorophylls and total carotenoids, the compounds were extracted with 90% methanol. Briefly, 2 mL of samples was centrifuged, and the pellet was suspended in 2 mL of 90% methanol and incubated at 70 °C for 5 min. The concentrations of chlorophylls and total carotenoids were measured according to the equations given by Lichtenthaler [28].

All spectrophotometric determinations were carried out by a Metter Toledo UV5 spectrophotometer and all analyses were performed at least in triplicates. The results are presented as means and standard deviations derived from three biological replicates.

### 2.4. Calculations

To estimate the maximum specific growth rate (μ_max_, d^−1^) during the exponential phase the following equation was applied:x = x_ο_ × e^(μmax × t)^(1)
which was converted into the linear equation:lnx = lnx_o_ + μ_max_ × t(2)

The biomass yield coefficient on glycerol (Y_x/s_) was estimated by the equation:Y_x⁄s_ = (x_max_ − x_o_)/(S_o_ − S_R_)(3)
where x_o_ (g/L) and x_max_ (g/L) represent the initial and the maximum biomass, respectively, t indicates the time of growth expressed in days (d), while S_R_ and S_o_ represent the concentrations of residual and initial glycerol in the growth media, given as (g/L).

### 2.5. Statistical Analysis

The statistical analysis was performed on analysis variance ANOVA, (one-way comparisons), conducted using SigmaPlot 12.0 software (Systat Software, Inc., San Jose, CA, USA). All data were tested for Normality (Shapiro–Wilk test) and for equal variance between treatments. The statistical analysis was based on Duncan’s pairwise multiple comparison procedure.

## 3. Results

### 3.1. Growth of S. quadricauda and C. vulgaris on Crude Glycerol under Heterotrophic and Mixotrophic Conditions

#### 3.1.1. Heterotrophic Conditions

To investigate the effect of crude glycerol on the heterotrophic growth of *S. quadricauda* and *C. vulgaris*, the microalgae were cultivated in a medium containing crude glycerol as the main carbon source at an initial concentration of 10 g/L and peptone as the nitrogen source at a concentration corresponding to a C/N ratio of 20:1. After 12 days of cultivation, the remaining glycerol reached 1.22 g/L and 0.95 g/L for *S. quadricauda* and *C. vulgaris*, respectively. The growth curves of both microalgae are depicted in Figure 1, indicating that *S. quadricauda* was able to achieve higher biomass yields than *C. vulgaris*, which can be clearly observed already from day 4. An evaluation of the kinetic parameters of the cultivation, as described in Table 1 for *S. quadricauda* and Table 2 for *C. vulgaris*, showed that *S. quadricauda* exhibited a higher growth rate, with a maximum specific growth rate value (μ_max_) of 0.68 d^−1^, whereas *C. vulgaris* had a μ_max_ = 0.39 d^−1^. Accordingly, biomass production was higher for *S. quadricauda* compared to *C. vulgaris*; more specifically, biomass for *S. quadricauda* was estimated at 1.47 g/L and for *C. vulgaris* at 0.79 g/L at the end of cultivation (12 days). In *S. quadricauda*, the bioconversion of crude glycerol into biomass (Y_x/s_) was 0.16 g/g, while in *C. vulgaris*, the amount of biomass produced from crude glycerol assimilation (Y_x/s_) was equivalent to 0.07 g/g, verifying that *S. quadricauda* can utilize crude glycerol as a carbon source for biomass growth approximately two times more efficiently than *C. vulgaris.*

#### 3.1.2. Mixotrophic Conditions

When grown mixotrophically on crude glycerol and artificial light, *S. quadricauda* achieved higher growth rates compared to heterotrophic mode, while in the case of *C. vulgaris*, only a slight increase was observed under blue and white (control) light (Table 2). Comparing the growth kinetics and the biomass yield of the two strains, *S. quadricauda* was found to be able to grow more rapidly and achieve higher cell biomass concentration than *C. vulgaris* at all different light wavelengths tested (Figure 2). After 12 days of cultivation, the remaining glycerol in the culture medium was estimated to be 0.97 g/L (white), 0.23 g/L (blue), 0.12 g/L (red), and 0.22 g/L (yellow) for *S. quadricauda*, while in case of *C. vulgaris*, the respective values were 0.81 g/L (white), 0.24 g/L (blue), 0.18 g/L (red), and 0.15 g/L (yellow).

When grown under red light, *S. quadricauda* exhibited the highest specific growth rate (μ_max_ = 0.89 d^−1^), followed by yellow (0.87 d^−1^), white (0.84 d^−1^), and blue (0.80 d^−1^). Regarding the final biomass yield, *S. quadricauda* grown mixotrophically under white light (control) reached a concentration of 1.57 g/L. Following the same trend, the highest biomass yield was observed under yellow and red light (1.86 and 1.79 g/L, respectively), while blue light did not favor the biomass growth, leading to a biomass concentration of 1.52 g/L (Table 1). Statistically significant differences occurred when comparing white light values to red and yellow light, but not when comparing white light values to blue light (Figure 3). The highest biomass yield as function of the remaining substrate (Y_x/s_) was reported in cultivation under yellow light with a value of 0.18 g/g, followed by red (0.17 g/g), white (0.16 g/g), and blue light (0.15 g/g).

Contrary to the results obtained for *S. quadricauda* where yellow and red light favored the cells proliferation and the final biomass yield, *C. vulgaris* achieved the highest specific growth rate when grown mixotrophically under blue (μ_max_ = 0.45 d^−1^) and white (0.43 d^−1^) light, followed by red (0.41 d^−1^) and, finally, yellow light (0.38 d^−1^). White light also led to the highest biomass concentration in *C. vulgaris* (1.34 g/L), followed by blue (0.90 g/L), red (0.82 g/L), and yellow light (0.76 g/L) (Table 2). All differences between the control (white light) and the treatments were statistically significant (Figure 3). The biomass production per gram of crude glycerol in *C. vulgaris* was highest under white light (Y_x/s_ = 0.13 g/g), followed by blue (0.08 g/g) and, finally, by red and yellow light (0.07 g/g) (Table 2).

### 3.2. Biochemical Composition of the Microalgal Cells

#### 3.2.1. Heterotrophic Conditions

The results obtained when using crude glycerol as the main carbon source in the growth medium for the production of metabolites (lipids, proteins, and carbohydrates) in *S. quadricauda* and *C. vulgaris* under heterotrophic conditions are shown in Table 1 and Table 2. For both microalgal strains, proteins constituted the major fraction of the cellular biomass, reaching a value of 44.22% of dry cell weight for *S. quadricauda* and 57.23% for *C. vulgaris*. The lipid content of *S. quadricauda* biomass was 14.17% of dry cell weight, which is slightly higher than that of *C. vulgaris* (11.23%), while carbohydrate content was 19.21 and 9.18% for *S. quadricauda* and *C. vulgaris*, respectively. These results are in accordance with previous studies in the literature [29,30,31]. The differences in production of carbohydrates between *S. quadricauda* and *C. vulgaris* can be attributed to the different metabolic pathways that are employed by each species for the assimilation of carbon source. The species-specific accumulation of carbohydrates, proteins, and lipids under the same growth conditions has been reported in the literature [32].

Regarding pigments (total chlorophylls and total carotenoids) under heterotrophic conditions, *C. vulgaris* had a content of 0.96% ± 0.11% of total chlorophylls and 0.22% ± 0.01% of total carotenoids, and *S. quadricauda* 0.42% ± 0.12% of total chlorophylls and 0.17% ± 0.07% of total carotenoids. These values are generally lower compared to their typical content (1.5–2.5% chlorophylls and 0.4–0.5% carotenoids), which can be explained by the fact that under heterotrophic conditions, microalgae tend to synthesize less photosynthetic pigments since the sole growth mechanism is switched from photosynthesis to heterotrophy, where pigments do not contribute to metabolism [33].

#### 3.2.2. Mixotrophic Conditions

Mixotrophic cultures of both *S. quadricauda* and *C. vulgaris* consisted of cells having a higher lipid content (%) than those in heterotrophic mode, and this was observed at all four light wavelengths tested with statistically significant differences. For *S. quadricauda*, the lowest lipid content was observed under white light and corresponded to 16.98% of dry cell biomass, while the values for red and blue light were very similar (20.65% and 20.12%, respectively), generally showing no significant differences compared to the white light (control) (Figure 4). In case of *C. vulgaris*, the lowest lipid content of the biomass was observed under both white and red light (15.16 and 15.88%, respectively), while the highest was achieved under blue light (26.47%), with a statistically significant difference between the values obtained under white and blue light (Figure 4).

Protein content (%) in mixotrophic cultures of *S. quadricauda* under white light exhibited only a slight increase (44.22% of dry cell biomass) compared to heterotrophy (45.18%), with the difference being not statistically significant. Comparing the various light wavelengths, treatment under white light exhibited the highest protein content followed by blue (40.32%), yellow (31.23%), and red (28.22%) light, all showing statistically significant differences from the white light (control) (Figure 4). As far as *C. vulgaris* is concerned, the maximum protein content was observed in red light (63.9%), a value which was higher than that appeared in heterotrophic conditions (57.23%). Treatment under white light led to cells with the lowest protein content (34.55%), while cells consisted of 53.12% and 54.78% of protein under blue and yellow light, with no statistically significant variations observed.

Carbohydrate content (%) in the microalgal biomass during the mixotrophic growth of *S. quadricauda* under white light was higher (23.09% of dry cell biomass) than that obtained in heterotrophic mode (19.21%) (Table 1); however, the values were lower under blue (14.16%), yellow (14.34%), and red light (16.76%). However, statistical analysis indicated no statistically significant differences between mixotrophic conditions with different wavelengths (Figure 4). The opposite trend was observed in *C. vulgaris,* where the carbohydrate content in mixotrophic cultures under white light (7.9%) was lower than in heterotrophic mode (9.18%). Under red and yellow light, the values were higher than in heterotrophic mode, reaching 18.13 and 20.34%, respectively, both demonstrating statistically significant differences when compared with white light.

Light quality had a diverse effect on the pigment content (total chlorophylls and total carotenoids) of *C. vulgaris* and *S. quadricauda*; the total chlorophylls and carotenoids content ranged between 0.58–0.80% and 0.22–0.27%, respectively, for *C. vulgaris* and 0.48–1.17% and 0.17–0.44%, respectively, for *S. quadricauda*. Both microalgal species had a higher pigment content under white and red lights compared to blue and yellow. Nevertheless, as in case of the heterotrophic growth, the pigments content was lower compared to the to their typical content under phototrophic conditions. It seems that also under mixotrophic mode, the heterotrophic metabolism had a stronger contribution on growth and, therefore, photosynthetic pigments were biosynthesized less than in the phototrophic mode.

### 3.3. Morphological Observations of the Cells

#### 3.3.1. Phenotype of *S. quadricauda* under Heterotrophic and Mixotrophic Conditions

*Scenedesmus* as genus is known for its extreme plasticity on its phenotype driven by environmental conditions [34]. The single-cell form of *Scenedesmus* spp. is more common in nature and in wild-type strains, and it is the life form that dominates in low initial cell density [35,36]. Usually, *Scenedesmus* consists of 4, 8, 16, or 32 cells arranged in a row, owning four spines, which are typical for some species and are considered to be important for cell buoyancy in water column. In this study, microscopic observation of *S. quadricauda* cells grown under mixotrophic conditions revealed the existence of two- or four-cell colonial structures (coenobia), consisted of lemon-shaped cells with a honeycomb appearance, typically arranged in a row (Figure 5), as previously reported in the literature [37]. *S. quadricauda* is typically formed as four cells arranged in a row. The coenobia are not motile and have four spines. The observations showed that the morphology of the cells grown in mixotrophic conditions was affected by the different light wavelength. More specifically, the number of the cells (two or four) that constituted the coenobia, as well as the average size of the two- or four-cell structure among the different light colors treatment, varied. The largest four-cell colonies (23.87 ± 1.37 μm) were produced during mixotrophic growth under red light treatment, followed by white (22.25 ± 0.8 μm) and yellow light (20.31 ± 0.3 μm) (Figure 6). In case of cultivation under blue light, *S. quadricauda* formed mainly two-cell colonies and, so, the average size was estimated by the measurement of these structures (8.80 ± 1.1 μm).

#### 3.3.2. Phenotype of *C. vulgaris* under Mixotrophic Conditions

The species of *Chlorella* are characterized as being small and spherical to ellipsoidal cells, with an average cell size of 5–10 μm. Contrary to *S. quadricauda*, *Chlorella* species are not characterized by plasticity in their morphology [38], as was also verified in the present study. Our results, as depicted in Figure 7, showed that in mixotrophic cultures, *Chlorella* cells appeared to be smaller than the typical size of the genus previously reported in the literature (approximately 10 μm) [38], probably due to the wavelength of the light source that was applied and the initial crude glycerol concentration that was supplied (10 g/L). Previous studies reported that lighting affect several some metabolic pathways related to cell size [16]. An evaluation of cell size under treatment with different light wavelength showed small differences (Figure 8), with bigger cells (4.65 ± 0.5 μm) to appear under cultivation with white light treatment and smaller cells (3.23 ± 0.2 μm) to appear mostly under yellow light.

## 4. Discussion

Developing novel, efficient strategies for optimizing microalgal growth conditions and metabolite production by utilizing low-cost, widely available industrial byproducts, and residual streams as fermentation substrates is of great interest. Utilization of crude glycerol that is obtained after biodiesel production has been gaining more and more interest as a carbon source for microalgae cultivation [6]. One step further involves combining crude glycerol with artificial light in mixotrophic cultures in order to increase the cell biomass yield and the synthesis of high-value products, which is a promising approach. LEDs are considered one of the most effective and accurate energy sources, as they can be easily tuned to emit the specific wavelength required for photosynthetic reactions. To investigate the growth and metabolite production of *S. quadricauda* and *C. vulgaris* under mixotrophy in this study, different wavelengths of monochromatic light were used, namely blue, red, and yellow, while white light was used as a control. In parallel, heterotrophic cultures using crude glycerol as the sole carbon source were run and evaluated.

The results showed that, although with lower growth rate than in mixotrophic conditions, both *S. quadricauda* and *C. vulgaris* were able to utilize crude glycerol as the sole carbon in heterotrophic conditions. These results are important, since previous studies in *C. vulgaris* with 0.5 g/L crude glycerol reported that no significant growth of microalgae was observed under heterotrophic conditions, despite the apparent consumption of nitrogen and carbon sources in the culture [39]. More specifically, authors explained that the biomass produced under heterotrophic conditions was 0.176 g/L in nitrogen replete and 0.1506 g/L in nitrogen deplete, while under mixotrophic conditions, the biomass was higher, reaching 0.37 and 0.238 g/L under nitrogen sufficiency and deficiency, respectively [39]. The higher cell concentration observed under mixotrophic conditions is in accordance with the results obtained in this study. Mou et al. (2022) obtained similar findings, demonstrating that microalgae growth was both better in the mixotrophic mode than in the heterotrophic mode [40]. Regarding *S. quadricauda,* though there has been no research on its growth on glycerol, when the microalga was grown heterotrophically on xylose with initial concentration of 4 g/L for 8 days culture, the maximum biomass production was above 0.44 g/L, which is more than three times lower than the biomass produced in the current study. Rai et al. (2016) cultivated *Senedesmus* sp. under mixotrophic conditions using glycerol at concentrations varying from 0% to 5% (*v*/*v*), observing a noteworthy enhancement in both growth and lipid accumulation. The biomass of *Scenedesmus* sp. increased by up to 10 g/L glycerol, reaching a recorded value of 0.414 g/L, which was approximately twice as much as the growth achieved through photoautotrophic cultivation (0.223 g/L) [41].

Regarding mixotrophic cultures, our results showed that *S. quadricauda* achieved the highest maximum specific growth rate in red light, the highest biomass production and biomass yield per g of glycerol consumed in yellow light, while *C. vulgaris* grown under mixotrophic conditions at 10 g/L of crude glycerol achieved the highest maximum specific growth rate in blue light, the highest biomass production and biomass yield per gram of glycerol consumed in white light. A study conducted with *C. vulgaris* and *Scenedesmus obliquus* consortium in mixotrophic conditions under different light wavelengths revealed that cool white light produced the highest amount of biomass and led to a higher maximum specific growth rate, followed by blue [42], which was also observed for *C. vulgaris* but not for *S. quadricauda* in this study. The results are in accordance with another work reporting that warm white light with an intensity of 80 mol/m^2^/s was revealed to be the ideal light for biomass synthesis of *C. vulgaris* [43]. On the contrary, the study conducted by Hultberg et al. (2014) reported that *C. vulgaris* was exposed to monochromatic light at six different wavelengths and showed that cultures that were exposed to yellow, red, and white light reached the log phase earlier than the other treatments and *C. vulgaris* cultures grown under yellow light, followed by white and red light, showed increased biomass percentages [44].

The effect of lighting was evident in the observed morphological differences, particularly in *S. quadricauda*, which displayed a different phenotype in response to various light treatments. *Scenedesmus* sp. exhibited high plasticity and it was greatly affected by fermentation conditions and, so, it was possible to evaluate the differential behavior of the cells depending on the wavelength under microscopical observation. The results are very important, since, until now, the existent knowledge obtained concerning the different phenotypes caused by the variety of the environmental conditions (i.e., light intensity, temperature) usually focuses on more or less uni-cell cultures [45]. In the current study, it was proven that only blue light caused the cells to create two-cell coenobia of a lesser size, while white, yellow, and red illumination promoted the formation of larger cells that created typical four-cell structures. Differences in the number of cells observed in microalgal coenobia may depend on many different factors and are indicative of the distinct metabolic pathways employed by the cells when grown under different light wavelengths [32]. Throughout the literature, the effect of cultivation factors, such as the type of carbon source, on coenobia formation is not well understood in *Scenedesmus* genus and, thus, it is not easy to provide an explanation. Since it has been suggested that during coenobia formation, increased energy resources are required by the cells, it can be assumed that white, yellow, and red light promote a faster cell division than blue light, which is in accordance with the lower growth rate and biomass productivity observed in this case.

Apart from the mixotrophic mode, the impact of different wavelength light sources has been also investigated in phototrophic cultivation of *C. vulgaris* [46], showing that shifting the wavelength from red to blue resulted in a 30% increase in the dry weight per cell of *C. vulgaris*. Blue light greatly boosted cell size, but red light resulted in small cells with active divisions [46], which is in accordance with the results of the present study, where the cells of *C. vulgaris* under blue light were bigger than those under red light. It has been previously reported that blue light increases the concentration of chlorophylls in microalgae cells, which are reordered in order to enhance their presence in the light-harvesting complex and reaction centers, hence increasing the size of the photosystem II (PSII). Microalgae adaptation to light conditions entails the operation of photoreceptors, which adjust these effects. Different photoreceptors (cryptochromes) act on the blue visible spectrum region in microalgae. This type of photoreceptor of the genus *Scenedesmus obliquus* work by activating the synthesis of chlorophylls in response to the intensity and profile of radiation received [47,48]. Τhe results of this study showed that blue light boosted the accumulation of lipids in *S. quadricauda* and, more profoundly, in *C. vulgaris*, while it also favored μ_max_ and carbohydrate synthesis in *C. vulgaris*, but not final biomass concentration and conversion of glycerol to cell biomass (Y_X/S_). Throughout the literature, blue light illumination has been reported to fail to promote *C. vulgaris* growth [42], and lead to low biomass production in comparison to red and white lighting in *A. protothecoides* [8] and *Auxenochlorella pyrenoidosa* [49]. Moreover, another study demonstrated that white and red light supports higher growth rates and biomass productivity in both *C. vulgaris* and *S. quadricauda* [50]. On the contrary, blue light has been reported to lead to a higher average rate of biomass production than the red or yellow illumination of *S. quadricauda* [51], which is not in accordance with the results of this study. The contradictory results regarding the effect of blue light observed in *S. quadriquda* and *C. vulgaris*, as well as within the different studies in the literature, can be attributed to the fact that that shorter wavelengths, such as blue light, deliver more energy for photosynthesis; however, they are often prone to cause photo-inhibition [49]. On the contrary, red light has a longer wavelength, which may mitigate the effect of photo-inhibition. Furthermore, the fact that different chlorophylls and related pigments found in microalgae exhibit the highest absorption peaks either in the red or blue regions of the light spectrum might also explain the differences between the various strains in response to the different light wavelengths. Last but not least, it has been reported that the content of *C. vulgaris* chlorophyll is strongly reliant on light wavelength (blue and red light, for example, are able to trigger the production of high and low levels of chlorophyll, respectively), thus corroborating the idea that each wavelength has a minimum intensity threshold for growth [52].

The production of valuable metabolites (lipids, proteins, and carbohydrates) accumulated in cells is an important parameter when optimizing microalgal fermentation processes, since these compounds have been found to account for approximately 61.5% of the ash-free biomass and this value is common in diatoms, green algae, as well as cyanobacteria [53]. Several studies have shown that the composition of the produced biomass varies greatly depending on the microalgal strain and the culture conditions (e.g., trophic mode, nitrogen deficiency, light intensity and light spectra, temperature, etc.). For example, in controlled conditions of growth under lighting, the protein content of *Chlorella* strains has been reported to range from 7 to 60%, the lipids portion from 3 to 29%, and the carbohydrates from 8 to 63% of dry biomass, whereas the corresponding values for *Scenedesmus* strains were 17 to 50%, 12 to 50%, and 6 to 52%, respectively [54], which are in accordance with the findings of the present study. Our results showed that both *S. quadricauda* and *C. vulgaris* are able to alter their biochemical pathways and, thus, the metabolite accumulation in response to different light wavelength. *S. quadricauda* achieved the highest biomass protein content in white light, the highest lipid content in blue and red light and the highest carbohydrate content in white light. The lowest values in biomass metabolite content were obtained for proteins in red, lipids in white, and carbohydrates in blue and yellow light. *C. vulgaris* achieved the highest biomass protein content in red light, the highest biomass lipid content in blue light and the highest carbohydrate content in yellow light. The lowest biomass content of proteins and carbohydrates was observed in white light and lipids in white and red light. When comparing the accumulation of lipids between heterotrophic and mixotrophic conditions, it is profound that the presence of light promotes lipid synthesis, which has been also reported in the literature [24]. Actually, under light, microalgae tend to produce higher total lipids and biomass due to the energy and coenzyme (NADPH) provided by photosynthesis, which might facilitate the expression of acetyl-CoA carboxylases to stimulate the synthesis of fatty acids. *S. quadricauda* lipid content has been shown to be significantly improved when growing mixotrophically (33.1% of dry cell weight) as compared to heterotrophical mode of cultivation (14–28%) [24], which was also observed in this study. Since it is known that high lipid content in microalgae is usually achieved at lower growth rates and/or stressful settings [55] and the wavelength of the lighting influences yield output and lipid content in microalgae, it can be assumed that the variable application of LED lights may stress the culture and lead to further lipid accumulation [56]. In the present study, blue light boosted lipid accumulation in both *C. vulgaris* and *S. quadricauda*, while similar results have been obtained with green lighting enhanced the formation of lipids [14]. Apart from lipid content, the wavelength of light source also affects accumulation of proteins and carbohydrates in microalgal cells; *C. vulgaris* and *S. obliquus* have been reported to synthesize highest protein content under cool white light followed by amber light, while amber light produced the highest carbohydrate content [42]. White light was also found to favor protein and carbohydrate synthesis in *S. quadricauda* but not in *C. vulgaris* in this study. Summarizing the above results, it is profound that light wavelength in mixotrophic cultivation mode greatly affects not only the growth of microalgae but also the accumulation of metabolites.

## 5. Conclusions

In the present study, it was shown that, despite exhibiting different metabolic behavior, both *S. quadricauda* and *C. vulgaris* were able to grow and produce valuable metabolites in 10 g/L crude glycerol, in either heterotrophic or mixotrophic conditions. Biomass yield, maximum specific growth rate, and lipid accumulation were higher in mixotrophic mode for both microalgal strains. In mixotrophic conditions, cell growth was obviously higher in *S. quadricauda* compared to *C. vulgaris*, while the light wavelength greatly affected the cell morphology and phenotype, the growth rate, and the production of biomass and metabolites for both microalgae, potentially regulating the activity of different metabolic pathways. As concluded, in addition to *C. vulgaris*, *S. quadricauda* is a promising microalga, particularly regarding lipid accumulation using crude glycerol under mixotrophic conditions.

## Figures and Tables

**Figure 1 foods-12-03068-f001:**
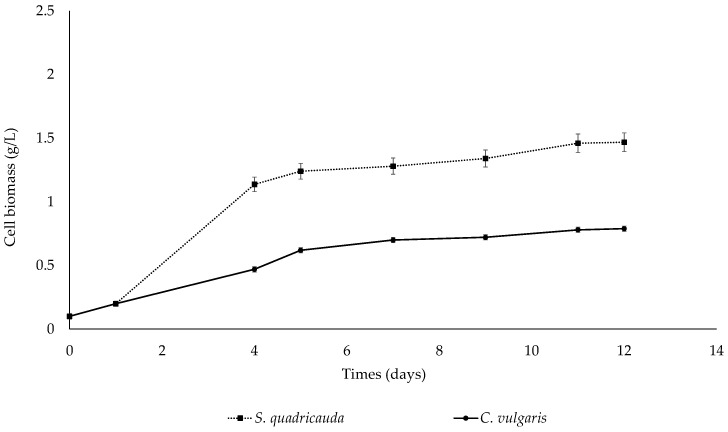
Kinetic growth of *S. quadricauda* and *C. vulgaris* under heterotrophic growth conditions in BBM medium with 10 g/L crude glycerol concentration as a carbon source, peptone as a nitrogen source, and a C/N ratio of 20:1.

**Figure 2 foods-12-03068-f002:**
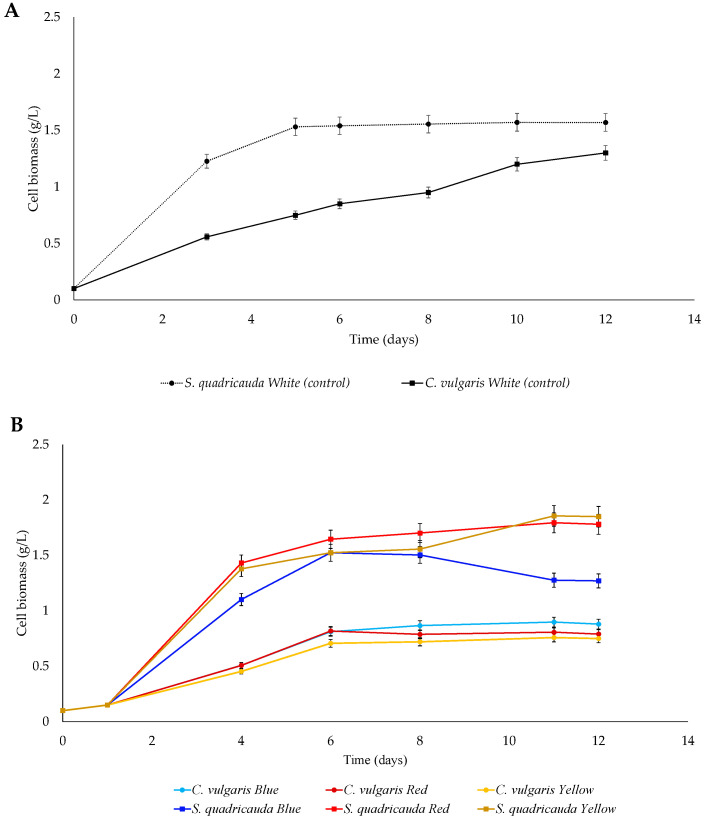
Kinetic growth of *S.quadricauda* and *C. vulgaris* under mixotrophic growth conditions in BBM medium with (**A**) white (control) and (**B**) blue, red, and yellow light, at an initial glycerol concentration of 10 g/L and peptone as a nitrogen source, with a C/N ratio of 20:1.

**Figure 3 foods-12-03068-f003:**
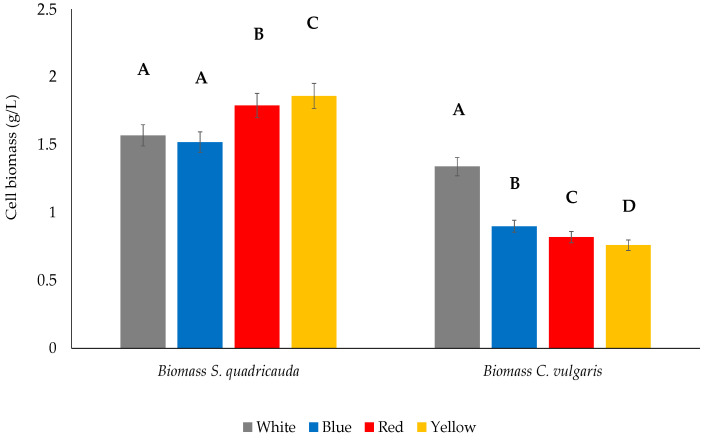
Biomass yield of *S. quadricauda* and *C. vulgaris* under mixotrophic growth conditions in BBM medium with 10 g/L crude glycerol as a carbon source, and peptone as a nitrogen source, with a C/N ratio of 20:1. The statistical analysis was performed with ANOVA, *p* < 0.05 of the pairwise comparisons between the different wavelengths. All comparisons were made with the white light specified as the control of the experimental procedure. The same capital letter denotes there are not any statistically significant differences between the means of pairwise comparisons of mixotrophic variables (white, blue, red, and yellow light). Different capital letters indicate statistically significant differences.

**Figure 4 foods-12-03068-f004:**
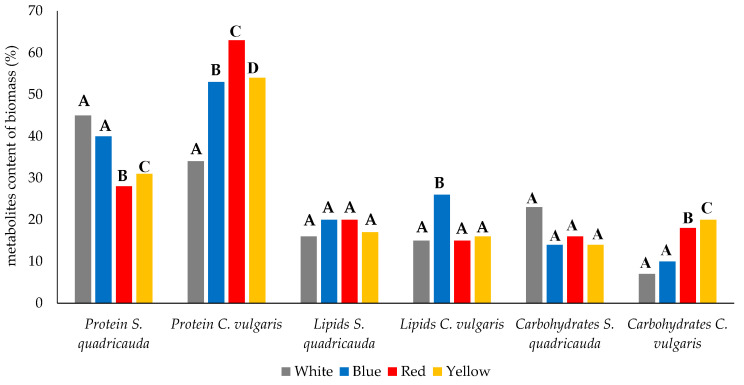
Metabolites (protein, lipids, carbohydrates) production of *S. quadricauda* and *C. vulgaris* under mixotrophic growth conditions in BBM medium with 10 g/L crude glycerol as a carbon source and peptone as a nitrogen source, with a C/N ratio of 20:1, under white, blue, red, and yellow light. The statistical analysis was performed with ANOVA, *p* < 0.05 of the pairwise comparisons between the different wavelengths. All comparisons were made with the white light specified as the control of the experimental procedure. The same capital letter denotes there are not any statistically significant differences between the means of pairwise comparisons of mixotrophic variables (white, blue, red, and yellow light). Different capital letters indicate statistically significant difference.

**Figure 5 foods-12-03068-f005:**
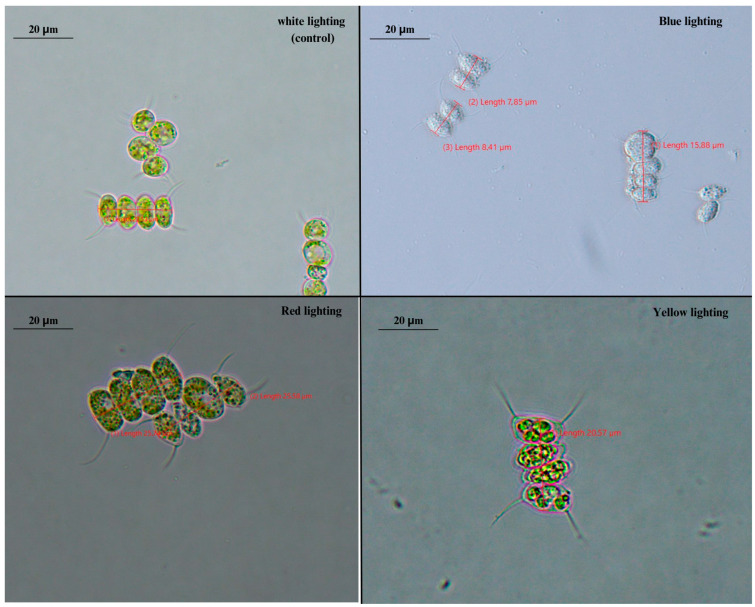
Phenotypical observation (100× magnification) of *S. quadricauda* when grown mixotrophically under different lighting wavelengths (white, blue, red, and yellow) and an initial glycerol concentration of 10 g/L on the steady phase.

**Figure 6 foods-12-03068-f006:**
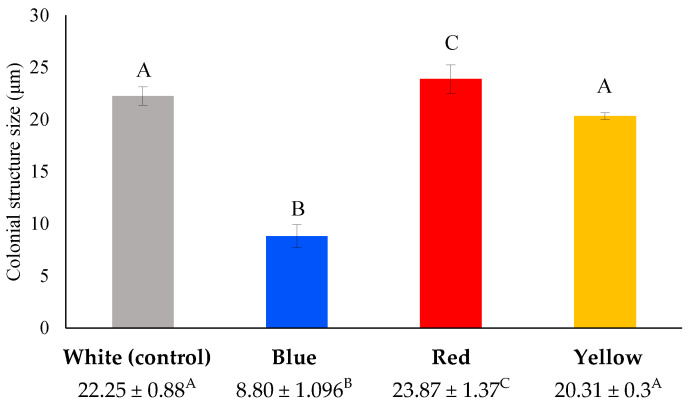
Phenotypical differences of *S. quadricauda* under mixotrophic growth conditions in BBM medium with 10 g/L crude glycerol as a carbon source and peptone as a nitrogen source at a C/N ratio of 20:1. The statistical analysis was performed with ANOVA, *p* < 0.05 of the pairwise comparisons between the different light wavelengths. All comparisons were made with the white light specified as the control of the experimental procedure. The same capital letter denotes there are not any statistically significant differences between the means of pairwise comparisons of mixotrophic variables (white, blue, red, and yellow light). Different capital letters indicate statistically significant differences.

**Figure 7 foods-12-03068-f007:**
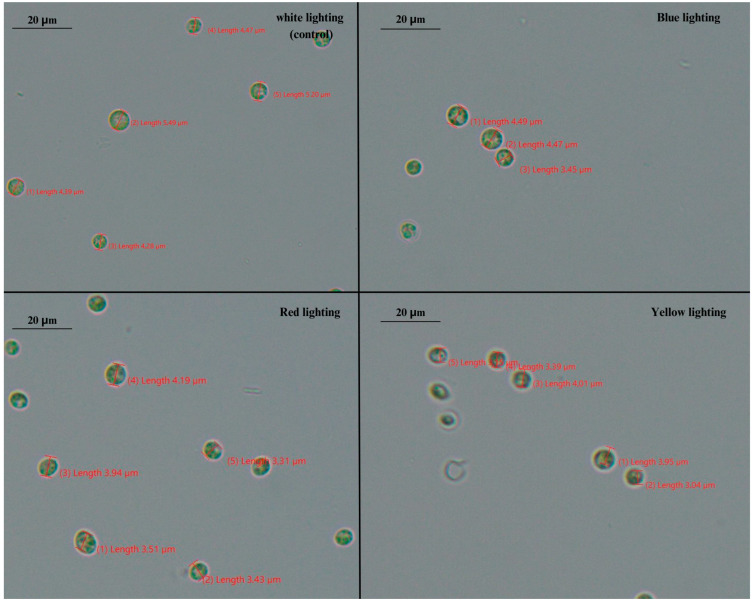
Phenotypical observation (100× magnification) of *C. vulgaris* when grown mixotrophically under different lighting wavelengths (white, blue, red, and yellow) and an initial glycerol concentration of 10 g/L.

**Figure 8 foods-12-03068-f008:**
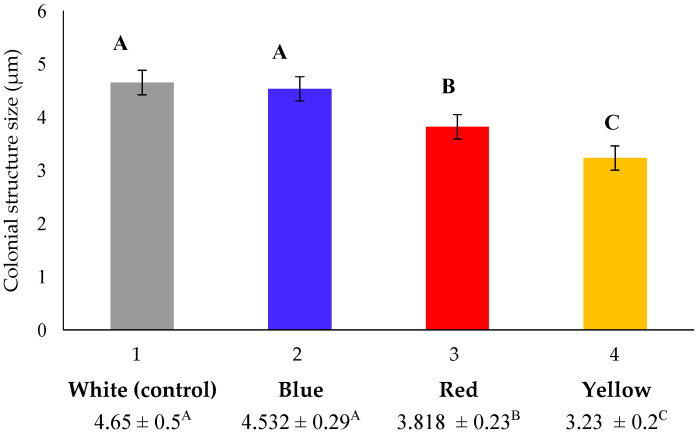
Phenotypical differences of *C. vulgaris* under mixotrophic growth conditions in BBM medium with 10 g/L crude glycerol as a carbon source and peptone as a nitrogen source at a C/N ratio of 20:1. The statistical analysis was performed with ANOVA, *p* < 0.05 of the pairwise comparisons between the different light wavelengths. All comparisons were made with the white light specified as the control of the experimental procedure. The same capital letter denotes there are not any statistically significant differences between the means of pairwise comparisons of mixotrophic variables (white, blue, red, and yellow light). Different capital letters indicate statistically significant differences.

**Table 1 foods-12-03068-t001:** Evaluation of μ_max_, biomass (X), proteins, lipids, carbohydrates, and Y_X/S_ produced during batch cultures of *S. quadricauda* under mixotrophic and heterotrophic mode of cultivation in BBM with 10 g/L crude glycerol and peptone (C/N ratio 20:1). Proteins, lipids, and carbohydrates are also expressed as a percentage of the dry weight of cell biomass. Data represent the average ± SD of three replicates (n = 3). Statistical analysis was performed with ANOVA, *p* < 0.05 between heterotrophy and mixotrophy. The same lowercase letter denotes there are not any statistically significant differences between the means of pairwise comparisons of heterotrophic and mixotrophic variables (white, blue, red, and yellow light). The difference in lowercase letters indicates statistically significant differences.

*Scenedesmus quadricauda*									
	μ_max_ (d^−1^)	X (g/L)	Y_X/S_	Proteins (g/L)	Proteins %	Lipids (g/L)	Lipids %	Carbohydrates (g/L)	Carbohydrates %
Heterotrophic mode									
	0.68 ^a^	1.47 ± 0.05 ^a^	0.16 ^a^	0.65 ± 0.03 ^a^	44.22 ^a^	0.206 ± 0.012 ^a^	14.17 ^a^	0.280 ± 0.022 ^a^	19.21 ^a^
Mixotrophic mode									
White (control)	0.84 ^b^	1.57± 0.04 ^b^	0.16 ^a^	0.71± 0.03 ^a^	45.18 ^a^	0.251± 0.008 ^b^	16.98 ^b^	0.361± 0.028 ^b^	23.09 ^b^
Blue	0.80 ^c^	1.52 ± 0.03 ^a^	0.15 ^a^	0.60± 0.03 ^a^	40.32 ^b^	0.304 ± 0.008 ^c^	20.12 ^c^	0.213± 0.018 ^c^	14.16 ^b^
Red	0.89 ^d^	1.79 ± 0.04 ^c^	0.17 ^a^	0.50± 0.05 ^b^	28.22 ^c^	0.358 ± 0.013 ^d^	20.65 ^d^	0.286 ± 0.023 ^a^	16.76 ^a^
Yellow	0.87 ^e^	1.86± 0.03 ^d^	0.18 ^a^	0.58± 0.04 ^a^	31.23 ^d^	0.316± 0.008 ^e^	17.89 ^e^	0.260± 0.024 ^a^	14.34 ^a^

**Table 2 foods-12-03068-t002:** Evaluation of μ_max_, biomass (X), proteins, lipids, carbohydrates, and Y_X/S_ produced during batch cultures of *C. vulgaris* under mixotrophy and heterotrophy (dark) in broth medium with 10 g/L crude glycerol and peptone (C/N ratio 20:1). Proteins, lipids, and carbohydrates are also expressed as a percentage of the dry weight of biomass. Data represent the average ± SD of three replicates (n = 3). The statistical analysis was performed with ANOVA, *p* < 0.05 of the pairwise comparisons between the different wavelengths and between heterotrophy and mixotrophy. The same lowercase letter denotes there are not any statistically significant differences between the means of pairwise comparisons of heterotrophic and mixotrophic variables (white, blue, red, and yellow light). The difference in lowercase letters indicates statistically significant differences.

*Chlorella vulgaris*									
	μ_max_ (d^−1^)	X (g/L)	Y_X/S_	Proteins (g/L)	Proteins %	Lipids (g/L)	Lipids %	Carbohydrates (g/L)	Carbohydrates %
Heterotrophic mode									
	0.39 ^a^	0.79 ± 0.06 ^a^	0.07 ^a^	0.45 ± 0.08 ^a^	57.23 ^a^	0.087 ± 0.008 ^a^	11.23 ^a^	0.071 ± 0.014 ^a^	9.18 ^a^
Mixotrophic mode									
White (control)	0.43 ^a^	1.34 ± 0.05 ^b^	0.13 ^a^	0.46 ± 0.07 ^a^	34.55 ^b^	0.201 ± 0.021 ^b^	15.16 ^a^	0.094 ± 0.014 ^a^	7.9 ^a^
Blue	0.45 ^a^	0.90 ± 0.07 ^a^	0.08 ^a^	0.48 ± 0.08 ^a^	53.12 ^a^	0.235 ± 0.018 ^c^	26.47 ^b^	0.090 ± 0.009 ^a^	10.11 ^a^
Red	0.41 ^a^	0.82 ± 0.04 ^a^	0.07 ^a^	0.52 ± 0.10 ^a^	63.9 ^a^	0.123 ± 0.017 ^d^	15.88 ^b^	0.148 ± 0.026 ^b^	18.13 ^a^
Yellow	0.38 ^a^	0.76 ± 0.04 ^a^	0.07 ^a^	0.41 ± 0.06 ^a^	54.78 ^a^	0.122 ± 0.008 ^e^	16.98 ^b^	0.152 ± 0.027 ^c^	20.34 ^a^

## Data Availability

Data is contained within the article.

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
