# Peer review of "Evaluation of Growth and Production of High-Value-Added Metabolites in Scenedesmus quadricauda and Chlorella vulgaris Grown on Crude Glycerol under Heterotrophic and Mixotrophic Conditions Using Monochromatic Light-Emitting Diodes (LEDs)"

_foods, 2023, doi:10.3390/foods12163068_

Round 1
Reviewer 1 Report
Some comments/suggestions:
1) Introduction: use more current references and bring other similar works. From this, show the difference between the current study in relation to what exists in the literature;
2) Highlight the novelty of the study;
3) Material and methods section: why did the authors not determine dry biomass in mg/L, for example?
4) Insert the statistical analysis section. Item 2.5;
5) Results and discussion section: improve the quality of images. The nomenclatures of the x and y axes are too small;
Reviewer 2 Report
I have few questions, please see PDF document

Please check briefly English language editing
Reviewer 3 Report
General comments:
Article titled “Growth kinetics and production of high value-added metabolites in Scenedesmus quadricauda and Chlorella vulgaris grown on crude glycerol under heterotrophic and mixotrophic conditions using monochromatic light-emitting diodes (LEDs)” deals with use of alternative C source – glycerol in heterotrophic and mixotrophic cultures of two fresh water microalgae species. It also studies effect of glycerol assimilation on the content of the basic biomass constituents (lipids, sugars, proteins and pigments) and effect of light source on cell morphology (under light microscope).
Line by line comments:
Title suggests kinetic parameters of glycerol uptake were calculated. However, no uptake rate was presented. Instead, you presented yields of biomass on glycerol as carbon source which is cumulative and not kinetic parameter (it does not depend on time). Please remove word kinetics from the title and use more appropriate word formation such as Biomass growth and production of high value-added metabolites…
In Chapter 2.1 Microorganisms and Culture Conditions it is stated: “illumination was provided by five LEDs (SMD type; 14.4 W per meter, 60 SMD LEDs per meter….”
Please clarify. So the illumination per each bottle was provided by only 5 LEDs equal to only 3 cm of LED tape per system?
How it was assured that this amount of LEDs provided 105 µmol/m2/s of PAR (that you claim) and how the light intensity was measured?
Also please add some characteristics of blue, red and yellow light you used in experiments, for example wavelengths of each colour.
Also in Results Section you mentioned that experiments lasted 12 days, please add the cultivation duration in this section as well.
In Chapter 2.2 Evaluation of cells morphology:
You used microscopy to assess morphology changes in different cultures. Have you also checked for eventual contaminations?
You used peptone, glucose and glycerol, was there any bacterial contamination during culturing process?
If not, please state clearly in this chapter.
In Chapter 3.1.1 Heterotrophic conditions
You state: “peptone was determined as the most suitable nitrogen source for the cultures of S. quadricauda and C. vulgaris when grown heterotrophically in crude glycerol (data not shown) and therefore was selected in our study”.
Instead of using “data not shown” phrase, please add in the ¨Supplementary material or Appendix the figure or table of these preliminary results clearly showing peptone advantage over nitrate/nitrate to support the selection of peptone in your study.
Figure 1. I suggest to add one more sub-figure to Figure 1 so you would have consistency with Figure 2 which is also divided to Figure 2a and 2b. Namely, you only give heterotrophic data for 10 g/L glycerol as C source, but not for the 10 g/L of glucose that you also used for culture maintenance. Adding to the Figure 1 the part of the growth with 10 g/L glucose (Figure 1a) would add information of the maximal performance under strictly heterotrophic growth. Then the Figure 1b would be already presented heterotrophig growth on 10 g/L glycerol. It would also look more informative and less empty.
Figure 1. also looks empty, y-axes scale goes to 2.5 g/L but the measured values do not exceed 1.5 g/L. Please rescale y-axis.
This also applies to Figure 2a.
Please state clearly how much of the initial glycerol added to the culture medium (10 g/L) was assimilated by the cells after 12 days of cultivation in all cases where you added glycerol.
In sentence “carbohydrate content was 19.21 and 9.18 % for S. quadricauda and C. vulgaris, respectively” how do you explain such a difference (2x higher) in sugar content at the same conditions for these 2 strains? Please add explanation to the main text.
In Table 1 and Table 2 you mention the statistical treatment of data using ANOVA. Please add Chapter dedicated to Data analysis and statistics to the bottom of the Materials and Methods section where you describe statistical methods and analysis applied and the software you used for the data treatment.
Figure 3. Please add standard deviation lines to the bar graph or explain the reason it was not depicted.
Discussion:
To the sentence “In the current study, it was proven that only blue light caused the cells to create 2-cell colonial structures of a lesser size, while white, yellow and red illumination promoted the formation of larger cells that created typical 4-cell structures” please add discussion on why this happens.
The sentence “Throughout the literature, blue light illumination has been reported to fail to promote C. vulgaris growth [34], and lead to low biomass production in comparison to red and white lighting in studies focused on A. protothecoides [8] and Auxenochlorella pyrenoidosa [41], while another demonstrates that white and red light supports higher growth rates and biomass productivity in both C. vulgaris and S. quadricauda [42]” is very long and hard to comprehend. Please reformulate. Also, please try to use shorter and more concise sentences where applicable.
The sentence “Since it is known than that high lipid content in microalgae is usually achieved at lower growth rates and/or stressful settings [48]and the wavelength of the lighting influences yield output and lipid content in microalgae, it can be assumed that the variable application of LED lights may stress the culture and lead to further lipid accumulation [49]” has some excess words (highlighted yellow).
The sentence “White light was also found to favor protein and carbohydrate synthesis in S. quadricauda but not in C. vulgaris in this study” lacks discussion. How do you explain this? Is it species specific or otherwise?
Reviewer 4 Report
Dear authors and editors,
The manuscript “Growth kinetics and production of high value-added metabolites in Scenedesmus quadricauda and Chlorella vulgaris grown on crude glycerol under heterotrophic and mixotrophic conditions using monochromatic light-emitting diodes (LEDs)" presents well-discussed and valuable data on the cultivation of two algal species that are widely used in human practise. The results obtained are important and have significance for the mass cultivation of algae to obtain high value-added products.
I would suggest including some points that may improve the manuscript:
1. In the last paragraph on page 9 it is written ……….Nevertheless, as in case of the heterotrophic growth, the pigments content was lower compared to the to their typical content under phototrophic conditions. – in my opinion to the is redundant;
2. Оn the 10th page it is written …………Scenedesmus as species is known for its extreme plasticity………... – Scenedesmus is actually a genus, not a species.
3. Оn the 10th page it is written ………Usually, Scenedesmus spp. consists of 4, 8, 16 or 32 cells arranged in a row, showing no motility despite owning 4 spiny bristles……..Here, first I would remove spp. after the genus name; second - 4 spines are typical for some species not for all of them and the spines have not to do with cell motility and they are important for buoyancy in water column and lastly - spiny bristles is not wide accepted term and I would just use spines.
4. Оn the 10th page it is written ….. In this study, microscopic observation of S. quadricauda cells grown under mixotrophic conditions revealed the existence of 2- or 4-cell colonial structures, consisted of lemon-shaped cells with a honeycomb appearance, typically arranged in a row (Figure 5) as previously reported in the literature [29]. S. quadricauda is typically formed as 4 cells arranged in a row. The cells are not motile and have 4 spiny bristles……..The colonial structures in Scenedesmus (and not only here) are actually called coenobia (pl.) or coenobium (sing.). And I would changethe last sentence here – The coenobia are not motile and have 4 spines.
5. In the last paragraph on page 11 it is written ………. Chlorella is a genus composed of multiple simple and tiny coccoid autosporic cells that are classified based on their morphological characteristics……..-
The genus cannot be composed of cells; it is composed from species for example; Autospores in Chlorella and all other coccal algae are cells for asexual reproduction and when the cell reproduces asexually it forms autospores.
6. When citing articles with more than two authors should be used et al. instead of and colleagues.
For example, Mou et al. (2022) obtained…. instead of Mou and colleagues (2022) obtained….
Round 2
Reviewer 3 Report
No additional requests/comments from my part.